# Long-Term Adherence to Continuous Positive Airway Pressure in Patients with Obstructive Sleep Apnoea Set Up in a Complete Remote Pathway: A Single-Centre Service Evaluation Project

**DOI:** 10.3390/jcm13102891

**Published:** 2024-05-14

**Authors:** Andras Bikov, Andrew Bentley, Balazs Csoma, Nicola Smith, Bryn Morris, Saba Bokhari

**Affiliations:** 1Manchester University NHS Foundation Trust, Manchester M23 9LT, UK; andrew.bentley@mft.nhs.uk (A.B.); csoma.balazs@semmelweis.hu (B.C.); nicola.smith@mft.nhs.uk (N.S.); bryn.morris@mft.nhs.uk (B.M.); saba.bokhari1@mft.nhs.uk (S.B.); 2Division of Immunology, Immunity to Infection and Respiratory Medicine, The University of Manchester, Manchester M13 9PL, UK; 3Department of Pulmonology, Semmelweis University, Tomo u 25-29, 1083 Budapest, Hungary

**Keywords:** obstructive sleep apnoea, continuous positive airway pressure, treatment adherence, telemedicine, COVID-19

## Abstract

**Background**: Continuous positive airway pressure (CPAP) is the first-line treatment for obstructive sleep apnoea (OSA). Maintaining adherence to CPAP in the long term is a clinical problem, and numerous factors have been identified that impact adherence. Although fully remote diagnostic and CPAP services were frequently utilised during the COVID-19 pandemic for patients with OSA, long-term adherence data have not been published. The aim of this service evaluation project was to describe the long-term adherence to CPAP. We also analysed factors that are associated with it. **Methods**: two-hundred and eighty patients diagnosed with OSA and set up on CPAP remotely during the first wave of the COVID-19 pandemic as part of routine clinical practice were analysed. **Results**: One-hundred and seven patients (38%) were fully adherent to CPAP at 24 months, determined by at least 4 h of usage on at least 70% of the days. Of the factors analysed, body mass index, disease severity, driving status and the presence of depression were related to long-term adherence (all *p* < 0.05). **Conclusions**: with the likelihood of future pandemics similar to COVID-19, our data provide evidence that fully remote pathways for management of patients with OSA can be designed and be sustainable with good long-term adherence.

## 1. Introduction

Obstructive sleep apnoea (OSA) is a common disease defined by repetitive partial or total collapse of the upper airways during sleep [1]. OSA may be associated with daytime sleepiness, impaired work and driving abilities and development of cardiovascular, metabolic and mental health disease. The gold standard diagnostic test is polysomnography; however, due to the high prevalence of OSA and limited access to inpatient sleep studies, various national guidelines support home sleep apnoea tests, with a caution that in the presence of significant cardiorespiratory disease, potential respiratory muscle weakness, hypoventilation, chronic opioid medication use, history of stroke or severe insomnia, polysomnography is recommended [2,3]. Continuous positive airway pressure (CPAP) is the first-line treatment for OSA [3]; however, it is an aerosol-generating procedure which had to be considered in the context of the COVID-19 pandemic [4].

COVID-19 placed an unprecedented burden on patients with OSA and sleep services worldwide [5]. Due to lockdowns, social isolation, partial or full closure of sleep services and concerns of viral transmission, many services utilised various remote diagnostic and treatment tools for patients with OSA [6]. This resulted in lower adherence to CPAP in 2020 compared to 2019, as reported by a multicentre study in the United Kingdom (UK) [6]. However, it is unclear if this was due to remote setups or other factors, most particularly mixed messages about the potential detrimental effect of CPAP on COVID-19 outcomes [7]. Nevertheless, our service has successfully adopted a fully remote diagnostic and CPAP service, where patients did not have any direct physical encounter with the service [8]. The adherence to CPAP (defined as at least 4 h of usage on at least 70% of the days) at 90 days was 41%, which was comparable to other tertiary services in the United Kingdom [6,9] and was similar to what was achieved in randomised controlled trials [10]. Whilst transformation to remote diagnostic and treatment services was necessary during the first waves of the pandemic, it was unclear if the level of adherence could be maintained in the long term.

Factors associated with adherence to CPAP are known to include OSA-related, comorbidity-related, socioeconomic and demographic factors [11]. The predictive value of these factors is weak and poorly reproducible, which prompts personalised treatment and encouragement by the clinical and technical staff when monitoring patients on CPAP. Nevertheless, as the fully remote pathway was unprecedented, the results of previous studies on adherence to CPAP, which were based on physical encounters, cannot be fully extrapolated to these settings. 

Therefore, the aim of the current service evaluation project was to determine the 2-year adherence to CPAP treatment of patients set up on the fully remote pathway. We also investigated factors related to adherence as well as long-term variations.

## 2. Materials and Methods

### 2.1. Subjects

In the beginning of the COVID-19 pandemic, the Regional Sleep Service at Wythenshawe Hospital, Manchester University NHS Foundation Trust, adapted local standard operating protocols for the remote diagnostic and treatment service. The routine clinical practice followed these protocols. According to these, adult patients with moderate to high pre-test probability for OSA (based on symptoms, body mass index and comorbidities) and those who were deemed to have capacity to make an informed decision (according to the Mental Health Act) were triaged to the remote diagnostic pathway. If the referral was suggestive for learning difficulties or sleep disorder other than OSA, such as parasomnia, narcolepsy or sleep-related movement disease, they were triaged directly to a consultant-led clinic to decide on the appropriateness of an inpatient sleep study. If following initial telephone encounter patients declined the remote diagnostic option, they were allocated to the face-to-face diagnostic waiting list. Those patients whose remote home cardiorespiratory polygraphy was suggestive for central sleep apnoea or hypoventilation syndrome were booked to a consultant-led telephone clinic and were set up on positive airway pressure as inpatients at our long-term ventilation service. If following the sleep test, during telephone appointments with the clinical staff, it was identified that the patient had difficulties with putting the interface on, they were scheduled for a face-to-face setup. Accordingly, patients who declined being diagnosed remotely, those with other sleep disorders than OSA, those with learning difficulties and those with difficulties putting the mask on did not participate in the service evaluation project. 

The NHS Health Research Authority decision tool (http://www.hra-decisiontools.org.uk/research/, accessed on 10 May 2020) deemed the project not to be research. The ethical approval requirements have further been discussed with the Institutional Research Office who confirmed on 16 September 2020 that the project does not require further NHS Ethics Review, HRA approval or registration with the Research Office. Data collection and processing were conducted according to the UK Data Protection Act 2018 and complied with the Caldicott principles, and data were not shared outside the care team. Therefore, as a service evaluation, the project did not require obtaining informed consent.

In the original service evaluation project, we analysed the results of the 300 patients (51 ± 13 years, 48% males, BMI 35/30–40/kg/m^2^) who were set up on CPAP between 20 May 2020 and 11 September 2020. The characteristics of the patients and the remote diagnostic and treatment pathway were described previously [8]. From the original database, we also excluded patients who died during the first 90 days following CPAP set up. Comorbidities were identified by accessing case notes, medications and primary care records at baseline, but they were not revisited following completion of this analysis. Driving status was evaluated during the telephone consultations according to the local practice to adhere to Driving and Vehicle Licencing Agency regulations. 

For the current analysis, we further excluded 20 patients. The reason for exclusion comprised death (10 patients), moving out of the area (7 patients), loss of the CPAP device (1 patient), escalation to adaptive servo-ventilation (1 patient) and non-invasive ventilation (1 patient) during the two-year follow-up period.

### 2.2. Cardiorespiratory Polygraphy

Patients received cardiorespiratory polygraphy via home delivery by a courier service, and they were supplied with written instructions and an online video outlining set up of the equipment. They returned the device the next day together with a completed questionnaire assessing their subjective sleep quality and quantity on the night of the analysis and an Epworth Sleepiness Scale (ESS). The questionnaire served as a quality control for the scoring physiologists, and the results were not evaluated in this service evaluation project. The cardiorespiratory polygraphy was performed with the Nox T3 Portable Sleep Monitor device (ResMed, Didcot, UK), and the tests were manually scored by trained sleep physiologists. Apnoea was defined as at least 90% reduction in the nasal flow lasting for at least 10 s. Hypopnoea was defined as at least 30% reduction in the nasal flow lasting for at least 10 s and associated with at least 3% drop in oxygen saturation. An apnoea–hypopnoea index (AHI) ≥ 5/h was sufficient to be diagnostic for OSA as patients had already been referred with suggestive symptoms or comorbidities [3]. 

### 2.3. CPAP Setup and Follow Up

Following a telephone consultation with a consultant in sleep medicine or a senior physiologist, patients were offered a CPAP device if the clinician felt that OSA was significantly responsible for their symptoms or comorbidities [3]. As a next step, we sent an information package and a template for an interface. An Airsense 10 device (ResMed UK) with the corresponding mask was delivered via a courier. Auto-set mode with 4–20 cmH_2_O minimum and maximum pressures included the default settings. All patients had a level 4 humidifier at baseline. Patients received a telephone call at 2, 7, 28 and 90 days to check their adherence and were provided with support by a physiologist. The patients were also encouraged to contact our helpline. If the patient was adherent to treatment at 90 days, they were booked for annual follow up. We applied various interventions, including changing settings, mask type, humidification, referral to psychologist or consultants to improve adherence. The usage was monitored with the Airview system (ResMed UK), and the patient was considered adherent if they used the device for at least 4 h on at least 70% of the days in the last 90 days.

This analysis was based on the extension of the former service evaluation project [8]. 

### 2.4. Statistical Analysis

JASP 0.14 (JASP Team, University of Amsterdam, Amsterdam, The Netherlands) was used for statistical analysis. Adherent and non-adherent groups at 2 years were compared with Mann–Whitney, Kruskal–Wallis and chi-squared tests. Factors associated with adherence were evaluated with linear and logistic regression analyses. Data are expressed as median/interquartile range. No a priori power or sample size calculations were performed, as this was a service evaluation project of routine clinical practice. A *p* value < 0.05 was considered significant.

## 3. Results

### 3.1. Comparison of the Adherent and Non-Adherent Groups at 24 Months

Thirteen patients (4.3%) never switched their device on, and 80 patients (26.7%) of the original 300 completely stopped using their device by 24 months. This included 10 patients who died during follow up, 7 patients who moved out from the area, 1 patient who lost their CPAP device and 2 patients whose CPAP was escalated to adaptive servo-ventilation and non-invasive ventilation. 

Analysing the remaining 280 patients, 124 (44%) were adherent at 28 days, 119 (43%) were adherent at 90 days, 100 (36%) were adherent at 12 months and 107 (38%) were adherent at 24 months.

Characteristics of adherent (at least 4 h of usage on at least 70% of the days in the last 90 days) and non-adherent patients were compared at 24 months (Table 1). Adherent patients had higher AHI, and the prevalence of drivers and the presence of depression was also higher in adherent patients (all *p* < 0.05).

### 3.2. Factors Associated with Adherence at 24 Months

Adherence was analysed both as a dichotomous variable (adherent vs. non-adherent) and as the average hours of usage at 2 years. The average hours of usage were related to the presence of anxiety (β = 0.26, *p* = 0.03), depression (β = 0.15, *p* = 0.02) and driving status (β = 0.11, *p* = 0.04), and there was a direct correlation with BMI (r = 0.17, *p* = 0.01) as well as AHI (r = 0.19, *p* < 0.01). Using logistic regression, adherence to CPAP at 24 months was associated with driving status (β = 1.08, *p* < 0.01), BMI (β = 0.03, *p* = 0.04), AHI (β = 0.02, *p* < 0.01) and the presence of depression (β = 0.683, *p* = 0.04). 

### 3.3. Changes in Adherence to CPAP over 24 Months

Adherence to CPAP was recorded at 28 days, at 90 days, at 12 months and at 24 months. Whilst 62 patients (22%) were continuously adherent (adherent at all time points), and 118 patients (42%) were continuously non-adherent (non-adherent at all time points), 100 patients (36%) showed variable adherence (evidence of adherence and non-adherence at different time points).

Comparing the three groups, continuously adherent patients were older. Patients who were continuously non-adherent were less likely to have depression, were less likely to be active drivers and had the lowest OSA severity (all *p* < 0.05, Table 2).

Figure 1 describes how the number of adherent and non-adherent patients changed over the follow-up period. At each time point, adherence improved or worsened in a considerable proportion of patients, indicating significant variability in adherence over the follow-up period.

### 3.4. Patient-Reported Reasons for Non-Adherence

We retrospectively reviewed the case notes to identify reasons for non-adherence in the 173 non-adherent patients at 24 months. The most common theme was mask intolerance, including claustrophobia and high leak in 32 patients (18.5%). Nine patients (5.2%) reported too-high pressures, nine (5.2%) dry mouth, two (1.2%) bloating, two (1.2%) facial pain and four (2.3%) salivation issues. 

Seventeen patients (9.8%) reported a decline in their somatic health and five (2.9%) worsening in their mental health as a reason for poor adherence. Back pain was reported in three patients (1.7%). Nasal issues prevented optimal CPAP use in three patients (1.7%) and four (2.3%) reported worsening dental health. 

Seventeen patients reported that they slept better when not using CPAP (9.8%), and five (2.9%) patients perceived no benefit. Six patients (3.5%) lost weight, two (1.2%) applied positional therapy, one patient (0.6%) opted for a mandibular advancement device and one for upper airway surgery (0.6%). 

Two patients (1.2%) stopped using CPAP completely after acquiring COVID-19, two (1.2%) were convinced that CPAP made their health worse, two (1.2%) did not accept their diagnosis and one (0.6%) used CPAP only when they felt it was necessary. Four patients (2.3%) tended to forget to put the mask on, and one (0.6%) did not resume CPAP after a night prayer. Three patients (1.7%) took off the mask involuntarily, one (0.6%) reported sleepwalking.

Seven patients (4.0%) reported changes in their personal circumstances, such as caring for relatives, another seven (4.0%) commented that CPAP disturbed their bed partner, including their children, and one patient (0.6%) told us that since a relationship break up snoring had not disturbed their new bed partner. Six patients (3.5%) reported worsening of their social situation as the reason for suboptimal adherence, including three (1.7%) who lost their homes and one (0.6%) who complained about an increase in electricity bills. 

## 4. Discussion

Face-to-face CPAP setups have additional benefits over remote setups. Firstly, they provide an opportunity to deliver further education by technical staff to patients on OSA and its impacts. Secondly, patients have the opportunity to try the CPAP device and different interfaces. They could experience immediate difficulties and the trained healthcare staff can provide immediate troubleshooting. In summary, a tailored personalised approach can be provided in terms of the right CPAP settings, humidification and choice of the best interface [12]. Taking into consideration that these opportunities are not available during remote setups, it may be expected that the short- and long-term adherence to CPAP during remote setup are inferior to face-to-face set up. Whilst a recent study has concluded this is not the case in the short term [9], to our knowledge, this is the first project evaluating the long-term adherence to CPAP in patients who were set up on a fully remote pathway.

There are only a limited number of studies that our results can be compared with [9,13,14]. A tertiary sleep service in the UK evaluated adherence to CPAP at 90 days in patients who were set up remotely and compared them to patients similarly set up during the COVID-19 pandemic but in a face-to-face setting. The adherence to CPAP at 90 days was comparable to our data (39% vs. 41%) [9]. Of note, patients in the face-to-face group had less severe OSA [9], which could be a potential bias, as our project has concluded that disease severity was associated with long-term adherence for patients in the remote pathway. A study in the United States compared patients who were set up either face-to-face or remotely; the latter group was divided into patients who received CPAP before and during the pandemic. The authors concluded that remote setups were associated with lower adherence (55% vs. 65%, remote vs. face-to-face, respectively) at 90 days; however, it did not matter if the remote setups happened before or during the pandemic [13]. Another US-based study evaluated a fully remote diagnostic and treatment setup [14]. The study recruited patients who were interested in taking part, thus introducing a potential selection bias. Adherence at 90 days was 65% [14]. The authors concluded that encouragement with a mobile telephone coach was associated with better adherence [14]. It is important to mention that two studies [9,14] used WatchPAT to diagnose OSA, which is a less specific method compared to cardiorespiratory polygraphy [15]. In addition, two studies [9,14] did not exclude comorbid sleep disorders, such as parasomnia or periodic limb movement disease. Therefore, residual sleep symptoms, potentially limiting CPAP adherence, cannot be excluded. Recently, Xu et al. published a non-inferiority trial which compared a fully remote diagnostic and treatment pathway to a traditional, face-to-face one [16]. The remote pathway was non-inferior in terms of adherence and treatment efficacy estimated by sleep quality; however, they only published adherence data at 3 months.

None of the three studies provided adherence data beyond 90 days [9,13,14]. Adherence to CPAP is known to decline with long-term usage [17]. However, our project highlighted that a significant number of patients show variable (both improving and worsening) adherence over time. This variation could be due to changes in disease-related or social circumstances; however, these were not recorded in the project. Apart from disease-related and socioeconomic factors and CPAP reimbursement initiatives, the different adherence rates between our current project and other studies could also be related to the information package provided to remote setups, as well as the method (i.e., video or phone call), quantity and quality of follow-up discussions. These factors were not evaluated in any of the studies and should be the focus of future research. 

Whilst the adherence rate in the current project is similar to other UK sites, it is lower than in other countries [13,18]. This could be explained by differences in reimbursing diagnosis and CPAP. However, they could also be a reflection of the lack of access to alternative treatments in the UK, such as upper airway surgery or a mandibular advancement device (MAD) during the COVID-19 pandemic, as some patients may have preferred these alternatives over CPAP [3,19]. Although we did not record if alternative treatments were offered to patients or whether they were available at the time of CPAP set up, approximately 5% of patients described changing OSA management as the reason for CPAP termination. Of note, surgery or MAD require direct patient contact and therefore were not routinely performed, especially at the beginning of the pandemic [20].

Interestingly, whilst it did not determine adherence at 90 days [8], disease severity was weakly associated with adherence at 24 months. The results are in line with other population-based studies from the UK [21], Spain [22] and France [23]. In addition, we found the patients with lower disease severity were unlikely to improve their adherence over the 2-year follow up. Our findings may imply that a remote CPAP setup is more feasible in patients with more severe disease. However, the project was not designed to propose cut-off values for AHI to predict long-term adherence. Of note, disease severity was determined by cardiorespiratory polygraphy, which tends to underestimate AHI compared to polysomnography. In addition, polysomnography may provide useful data about sleep architecture, such as sleep efficiency, which may predict non-adherence [24].

In line with the previous findings [25], driving status was associated with good long-term adherence. This is not surprising, as adherence to CPAP is pivotal for the continuation of a driving licence in sleepy patients with OSA in the UK. However, driving may also be an indicator for higher socioeconomic status. Patients living in more deprived areas have lower adherence rates to CPAP [26,27], and they also experience fewer benefits from novel technologies, including telemedicine [28]. Secondly, non-drivers may have difficulties in assessing face-to-face help or consumables if they live far from the sleep laboratory. The current project, however, did not analyse these factors in detail, which is one of the limitations. 

Surprisingly, we found that patients with depression were more likely to be adherent to CPAP in the longer term, whilst other recorded comorbidities did not impact adherence. The results on how depression is associated with adherence to CPAP are conflicting [11,29,30]. The other UK-based study by Meurling et al., which also investigated a remote CPAP pathway, did not find a significant association [9]. It is possible that patients with depression who were adherent to CPAP experienced significant improvement in their mental health symptoms and were keen to pursue it, thereby explaining our findings [31]. We did not assess their mental health in detail and, more importantly, whether CPAP resulted in an improvement in their mental health. Nevertheless, the direct relationship between better adherence and depression in the context of the pandemic warrants further research. 

We did not find a correlation between adherence at 2 years and age, gender or cardiovascular comorbidities. Of note, we noticed that patients who were continuously adherent tended to be older compared to those with continuous non-adherence or variable adherence. Gender and age are known to potentially affect adherence; however, a systematic review concluded that the results of the studies are contradictory; therefore, they need to be interpreted with caution [11]. As this was a service evaluation project, we did not investigate the relationship between age and CPAP preference in detail, as this could include complex socioeconomic factors which were not collected due to the nature of the project. Whilst cardiovascular risk prevention with CPAP is commonly discussed with the patients during their consultation, and this information may improve CPAP usage [11], less than 10% of our patients suffered from a cardiac disease; therefore, we could not conclude whether the presence of cardiovascular disease could be related to adherence to CPAP.

It is important to interpret our data with caution, as the patients were set up during the first wave of the COVID-19 pandemic. Apart from the restricted access to healthcare, patients may have been directly affected by COVID-19. Ten patients (3.3%) from the original cohort died and 10% reported a decline in their somatic health (potentially due to limited access to healthcare) as the reason for non-adherence. In addition, 2% reported non-adherence following contracting COVID-19. Furthermore, we noticed worsening socioeconomical status as a significant reason for CPAP termination. Unfortunately, COVID-19-related, direct and indirect factors were not monitored in this service evaluation project.

The project has further limitations. Firstly, we did not systematically record mask leak, residual respiratory events or residual symptoms in our project, which could all be potentially related to limited adherence. Secondly, only ESS was used to assess symptoms and was only undertaken at baseline. Apart from excessive daytime sleepiness, patients may complain about sleep initiation and maintenance insomnia, daytime tiredness, fatigue, memory and performance problems. These symptoms may relate differently to adherence than excessive daytime sleepiness. Thirdly, we did not record the patients’ marital or partnership status or if they shared their bed with others. Whilst social support is associated with adherence [32], while reviewing the reasons for non-adherence in this cohort, we found that sharing a bed can also be detrimental. Fourthly, whilst the project has evaluated a fully remote diagnostic and treatment pathway, it must be emphasised that patients had the opportunity to attend face-to-face diagnostic and treatment appointments. They also had the opportunity to attend face-to-face appointments during their follow up, especially with relieving COVID-19 related restrictions. We did not assess if these interventions and the presence of a supportive and responsive system for patients were associated with adherence. This was a single-centre service evaluation project; therefore, the data were not intended to be generalisable. Associated factors with adherence, or the lack of associations, therefore needed to be interpreted with caution.

Patients were recruited in the first year of the COVID-19 pandemic, and pandemic-specific factors may limit generalisability of the findings. Nevertheless, telemedicine has been applied more frequently in sleep medicine [33]. A recent meta-analysis has concluded that remote monitoring can successfully and effectively be applied in OSA [34]. Our findings support the premise that a fully remote approach can be delivered successfully in many patients with OSA. However, studies focusing on which patient would benefit from such an approach are still warranted.

## Figures and Tables

**Figure 1 jcm-13-02891-f001:**
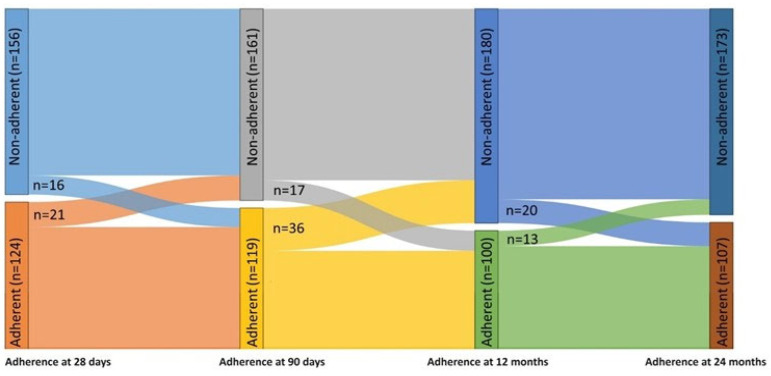
Sankey diagram demonstrating the changes in adherence over the follow-up period.

**Table 1 jcm-13-02891-t001:** Comparison of the adherent and non-adherent groups at 24 months.

	Adherent (*n* = 107)	Non-Adherent (*n* = 173)	*p* Value
Age (years)	52/43–60/	51/43–60/	0.807
BMI (kg/m^2^)	36/30–40/	34/30–40/	0.259
Gender (males%)	48	47	0.891
Chronic airway diseases (%)	16	18	0.777
Hypertension (%)	32	27	0.451
Ischaemic heart disease (%)	9	10	0.762
Cerebrovascular disease (%)	3	1	0.326
Atrial fibrillation (%)	5	5	0.807
Chronic heart failure (%)	3	4	0.732
Diabetes (%)	12	17	0.215
GORD (%)	12	15	0.391
Anxiety (%)	7	3	0.143
Depression (%)	21	12	**0.040**
No comorbidities (%)	18	16	0.606
ESS	11.5/8.0–15.0/	12.0/8.0–16.0/	0.340
AHI (events/hour)	42.0/27.0–60.5/	31.9/20.5–45.2/	**<0.001**
Driver (%)	90	75	**0.003**

Statistically significant difference (*p* < 0.05) is marked with bold text. Abbreviations: AHI—apnoea-hypopnoea index; BMI—body mass index; ESS—Epworth Sleepiness Scale; GORD—gastroesophageal disease.

**Table 2 jcm-13-02891-t002:** Comparison of the continuously adherent, continuously non-adherent and variable adherence groups.

	Continuously Adherent, *n* = 62	Continuously Non-Adherent, *n* = 118	Variable Adherence, *n* = 100	*p* Value
Age (years)	56/48–62/	51/40–61/	49/42–58/	**0.032**
BMI (kg/m^2^)	36/30–40/	34/30–40/	35/30–40/	0.524
Gender (males%)	44	47	49	0.793
Chronic airway diseases (%)	18	18	16	0.948
Hypertension (%)	31	28	29	0.949
Ischaemic heart disease (%)	5	10	12	0.335
Cerebrovascular disease (%)	2	1	3	0.460
Atrial fibrillation (%)	3	4	8	0.440
Chronic heart failure (%)	2	4	4	0.654
Diabetes (%)	13	19	11	0.191
GORD (%)	13	16	12	0.678
Anxiety (%)	3	6	5	0.426
Depression (%)	9	26	15	**0.013**
No comorbidities (%)	18	16	17	0.946
ESS	12.0/8.8–15.0/	12.0/8.0–16.5/	11.0/7.8–16.0/	0.824
AHI (events/hour)	41.4/24.0–59.8/	31.0/21.0–43.9/	38.9/23.0–54.0/	**0.019**
Driver (%)	92	71	85	**0.002**

Abbreviations: AHI—apnoea–hypopnoea index; BMI—body mass index; ESS—Epworth Sleepiness Scale; GORD—gastroesophageal disease. Statistically significant difference (*p* < 0.05) is marked with bold text

## Data Availability

The data presented in this project are available upon reasonable request from the corresponding author.

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
