# Peer review of "Long-Term Adherence to Continuous Positive Airway Pressure in Patients with Obstructive Sleep Apnoea Set Up in a Complete Remote Pathway: A Single-Centre Service Evaluation Project"

_jcm, 2024, doi:10.3390/jcm13102891_

Round 1

Reviewer 1 Report

Comments and Suggestions for Authors

The manuscript is generally well-written, but there are some points need some clarification

1- basis for selection sample size

2-authors should provide information about selected patients: age, gender, BMI

3- Did the patient sign a consent: the authors may add a copy of consent as supplementary data

4- the authors should state the centre where the study was performed

5- the authors should provide in discussion explanation why there is no significant difference among age ,CAD, heart disease, gender as factors to adherence

6- limitations to this study should be included.

7- further analysis to depression as a factor might be an interesting project that requires further investigation thus I suggest the authors could include future research to unanswered questions

Comments on the Quality of English Language

English is fine 

Author Response

Comment: 1- basis for selection sample size

Response: This was a service evaluation project; therefore, no a priori sample size calculations were made.

Comment: 2-authors should provide information about selected patients: age, gender, BMI

Response: Thank you. We added this information to the manuscript. Please, see section 2.1.

Comment: 3- Did the patient sign a consent: the authors may add a copy of consent as supplementary data

Response: Thank you very much for the comment. We confirm that the project was a service evaluation. The NHS Health Research Authority decision tool (http://www.hra-decisiontools.org.uk/research/) defined the study not to be research. The ethical approval requirements have further been discussed with the Institutional Research Office who confirmed on 16.03.2020 that the project does not require further NHS Ethics Review, HRA approval or registration with the Research Office.

The data collection and processing were fully adherent to the UK Data Protection Act 2018 and complied with the Caldicott principles. The data was not shared outside the care team, no visual or audio requirement was made and for the published data was anonymous. Therefore, the study did not require obtaining informed consent.

Comment: 4- the authors should state the centre where the study was performed

Response: This was a single centre service evaluation project conducted at the Regional Sleep Service at Wythenshawe Hospital, Manchester University NHS Foundation Trust. Section 2.1. has now been expanded to include this information.

Comment: 5- the authors should provide in discussion explanation why there is no significant difference among age, CAD, heart disease, gender as factors to adherence

Response: Thank you for the comment. We discussed this in the revised manuscript. Please, see 8th paragraph of Discussion.

Comment: 6- limitations to this study should be included.

Response: Thank you. Limitations were addressed in the 9th and 10th paragraph of the Discussion. We have expanded this section including a caution about generalisability of findings of a service evaluation project.

Comment: 7- further analysis to depression as a factor might be an interesting project that requires further investigation thus I suggest the authors could include future research to unanswered questions

Response: Thank you very much. This was included to Discussion, 7th paragraph.

Reviewer 2 Report

Comments and Suggestions for Authors

Dear Authors,

Your study on the long-term adherence to continuous positive airway pressure (CPAP) in patients with obstructive sleep apnea through a fully remote approach is fascinating. With the increasing utilization of telemedicine in sleep medicine, investigating CPAP adherence through remote prescription methods is crucial. The findings could enable physicians to manage obstructive sleep apnea entirely remotely during pandemics, for patients facing mobility challenges, or those lacking access to sleep laboratories with confidence. I believe that a key element contributing to the comparable adherence observed in your study was the presence of a supportive and responsive system for patients, which should be highlighted further in the discussion.

Author Response

Comment: I believe that a key element contributing to the comparable adherence observed in your study was the presence of a supportive and responsive system for patients, which should be highlighted further in the discussion.

Response: Thank you very much your supporting comments. This has been now discussed. Please, see Discussion, 10th paragraph.

Reviewer 3 Report

Comments and Suggestions for Authors

Comments to the authors

Thank you for inviting me to review the “Long-term adherence to continuous positive airway pressure in patients with obstructive sleep apnea set up in a complete remote pathway”. 

This a retrospective study of evaluation of adherence vs non-adherence at 24 months after delivering and following-up with CPAP treatment fully on telehealth. 

The article is very well written. It is very interesting, it provides useful information on long-term adherence and analysis of adherent vs non-adherent patients in a fully remote management. 

Only minor points are identified below:

Title: the title or the abstract should contain indication of the type of study design.

Introduction: this is well written. The background is nicely summarized; information on similar articles is provided; the aim is clear. 

One minor point is to describe the conditions when a home sleep apnea test is suggested based on the AASM (line 36).

I also suggest that the authors include the hypothesis after their aim. 

Methods:
Line 71 -> it is not clear what this exclusion criterion refers to “Patients who 71 did not have capacity”

Line 76-77: this sentence is also not clear “The excluded patients attended in-hospital polysomnography, in-hospital CPAP setup and the 77 long-term ventilation service, respectively.”

Line 89-90: how were these outcomes evaluated? “subjective sleep quality and quantity on the night of the analysis”

Line 95: “An apnoea-hypopnoea index (AHI) ≥ 5/h was diagnostic 95 for OSA”, please provide reference. 

Line 117: “Adherent and non-adherent groups at 2 years” would benefit from definition of adherence also here (not only in the abstract and introduction).

Results:

Line 124-130 would benefit from a graphical flowchart / figure showing the N of participants lost / those who were adherent, etc. 

Line 133: how was “prevalence of drivers” evaluated? This is not mentioned in the methods. Also, was depression revelaed by the medical history? Were the medical comorbidities reassess at 24 months or were they the ones extrapolated at baseline? 

Discussion: this is very well presented, with the critical speculation of the findings of the study within the context of the existing literature. Limitations are nicely acknowledged. 

Conclusions are supported by the current findings.  

Author Response

Comment: the title or the abstract should contain indication of the type of study design

Response: This was a service evaluation project which aimed to describe adherence to CPAP at 2 years following CPAP setup. We have modified the title and the abstract to clarify this. 

Comment: One minor point is to describe the conditions when a home sleep apnea test is suggested based on the AASM (line 36).

Response: Thank you. This has now been added.

Comment: I also suggest that the authors include the hypothesis after their aim. 

Response: This was a service evaluation project with an aim to describe adherence to CPAP at 2 years following CPAP setup. Therefore, no a priori hypothesis was formed, and no a priori sample calculations were performed.

Comment: Line 71 -> it is not clear what this exclusion criterion refers to “Patients who 71 did not have capacity”

Response: We were referring to the Mental Health Act. We have rephrased section 2.1.

Comment: Line 76-77: this sentence is also not clear “The excluded patients attended in-hospital polysomnography, in-hospital CPAP setup and the 77 long-term ventilation service, respectively.”

Response: Following the comments of the reviewers were significantly rephrased section 2.1. We hope that the current wording better explains the practice.

Comment: Line 89-90: how were these outcomes evaluated? “subjective sleep quality and quantity on the night of the analysis”

Response: The questionnaire served as a quality control for the scoring physiologists and the results were not evaluated in this service evaluation project.

Comment: Line 95: “An apnoea-hypopnoea index (AHI) ≥ 5/h was diagnostic 95 for OSA”, please provide reference. 

Response: Thank you. We clarified this. Please, see section 2.2.

Comment: Line 117: “Adherent and non-adherent groups at 2 years” would benefit from definition of adherence also here (not only in the abstract and introduction).

Response: This information was added.

Comment: Line 124-130 would benefit from a graphical flowchart / figure showing the N of participants lost / those who were adherent, etc. 

Response: We added a graphical abstract which summarises this information.

Comment: Line 133: how was “prevalence of drivers” evaluated? This is not mentioned in the methods. Also, was depression revelaed by the medical history? Were the medical comorbidities reassess at 24 months or were they the ones extrapolated at baseline? 

Response: Thank you for the question. Driving status and the implication of OSA on driving performance was evaluated during the telephone appointments according to the Driving and Vehicle Licensing Agency regulations in the UK. Comorbidities, including depression were identified by accessing case notes, medications, and primary care records at baseline; but they were not revisited following completion of this analysis. We have expanded section 2.1 with this information.